# The Potential of Bacilli-Derived Biosurfactants as an Additive for Biocontrol against *Alternaria alternata* Plant Pathogenic Fungi

**DOI:** 10.3390/microorganisms11030707

**Published:** 2023-03-09

**Authors:** Jesse John Sakiyo, Áron Németh

**Affiliations:** Department of Applied Biotechnology and Food Science, Faculty of Chemical Technology and Biotechnology, Budapest University of Technology and Economics, Műegyetem rkp. 3., 1111 Budapest, Hungary

**Keywords:** antifungal effect, biosurfactant, bacillus, sustainability, biocontrol

## Abstract

Fungal diseases caused by *Alternaria alternata* constitute a significant threat to the production and quality of a wide range of crops, including beans, fruits, vegetables, and grains. Traditional methods for controlling these diseases involve synthetic chemical pesticides, which can negatively impact the environment and human health. Biosurfactants are natural, biodegradable secondary metabolites of microorganisms that have also been shown to possibly have antifungal activity against plant pathogenic fungi, including *A. alternata* being sustainable alternatives to synthetic pesticides. In this study, we investigated the potential of biosurfactants of three *bacilli* (*Bacillus licheniformis DSM13*, *Bacillus subtilis DSM10*, and *Geobacillus stearothermophilus DSM2313*) as a biocontrol agent against *A. alternata* on beans as a model organism. For this fermentation, we describe using an *in-line* biomass sensor monitoring both permittivity and conductivity, which are expected to correlate with cell concentration and products, respectively. After the fermentation of biosurfactants, we first characterised the properties of the biosurfactant, including their product yield, surface tension decrement capability, and emulsification index. Then, we evaluated the antifungal properties of the crude biosurfactant extracts against *A. alternata*, both in vitro and in vivo, by analysing various plant growth and health parameters. Our results showed that bacterial biosurfactants effectively inhibited the growth and reproduction of *A. alternata* in vitro and in vivo. *B. licheniformis* manufactured the highest amount of biosurfactant (1.37 g/L) and demonstrated the fastest growth rate, while *G. stearothermophilus* produced the least amount (1.28 g/L). The correlation study showed a strong positive relationship between viable cell density VCD and OD600, as well as a similarly good positive relationship between conductivity and pH. The poisoned food approach in vitro demonstrated that all three strains suppressed mycelial development by 70–80% when applied with the highest tested dosage of 30%. Regarding in vivo investigations, *B. subtilis* post-infection treatment decreased the disease severity to 30%, whereas *B. licheniformis* and *G. stearothermophilus* post-infection treatment reduced disease severity by 25% and 5%, respectively. The study also revealed that the plant’s total height, root length, and stem length were unaffected by the treatment or the infection.

## 1. Introduction

In recent decades, the toxicity of chemically synthesised fungicides has become an increasing problem, so technologies that offer more sustainable, environmentally friendly solutions have come to the fore in all areas of industry [1]. Surfactants are molecules that can reduce surface tension and are used in various applications, including cleaning products, detergents, cosmetics, and agriculture. Synthetic surfactants have traditionally been used for these purposes, but there has been a growing interest in using surfactants produced by living organisms, particularly microorganisms, as a sustainable alternative. Biosurfactants have several advantages over synthetic surfactants, including being less toxic to the environment and health, biodegradability, and using renewable raw materials [2,3]. They have also been shown to possess antimicrobial activity, making them potential candidates for use in medicine and agriculture. However, the high production costs of biosurfactants have limited their widespread use [4].

Several research efforts are being conducted to identify novel bioprotective and antibacterial chemicals for technological and medical uses and for treatment-resistant diseases. However, it is difficult to determine the types of secondary metabolites a microorganism produces due to a lack of understanding of the molecular mechanisms underlying their production. Primary approaches often involve a variety of cultivation conditions and experimental assays before determining the types of secondary metabolites produced by a microorganism [5,6].

The increasing incidence of fungal infections is a rising health and economic problem for plants, people, and animals. *Alternaria alternata* is one of the most frequent pathogenic fungi owing to its vast host range and capacity to produce severe infections in plants and animals [7,8,9]. This fungus is known to cause leaf spots, fruit rot, and stem canker in crops, such as apples, tomatoes, and pepper. In addition, it has been reported as a causative agent of allergic bronchopulmonary mycosis in immunocompromised individuals [10,11,12].

Monitoring microorganism concentration during bacterial fermentations is one of the critical tasks and challenges of bioprocesses. Traditional methods involve taking and analysing samples off-line, which can be time-consuming and provide only discrete data points [12,13,14]. Standard techniques for detecting cell density during fermentation, such as optical density, turbidity, and dry cell weight, are often used to determine the total number of cells in a sample. However, the dilution plate (CFU-colony forming unit) and most probable number (MPN) techniques are routinely used for more exact counts of living cells. These methods are believed to be more accurate, although they are more labour-intensive and time-consuming than conventional processes. In-line sensors can provide nearly continuous measurements, which are helpful in process analytical technology (PAT) and quality by design (QbD) principles [15,16,17]. Capacitance methods are fast, capable of in-line measurement providing living cell numbers, and less sensitive to external influences. The Incyte capacitance sensor by Hamilton (Bonaduz, Switzerland) measures permittivity at several frequencies, resulting in a permittivity profile vs. frequency called dielectric spectroscopy; this allows for the almost continuous determination of living cell numbers [18]. At the same time, it also follows the conductivity, which may correlate to the pH changes in the broth mainly caused by primary metabolites.

Recent studies have shown that a biosurfactant extract produced from a *Bacillus* strain had significant antifungal action against *A. alternata* [19]. The extract inhibited the development of the fungus and reduced the severity of leaf spot symptoms in apple plants. In addition, the extract was non-toxic to plant cells and had no detrimental impact on plant growth and development. This study demonstrates the possibility of employing biosurfactant extracts from *Bacillus* spp. as a natural and efficient management method against pathogenic *A. alternanta* plant fungal infections [20,21,22,23].

The present study aims to produce and characterise biosurfactants from *Bacillus subtilis, Bacillus licheniformis*, and *Geobacillus stearothermophilus* and to investigate the antifungal properties of their biosurfactants against *A. alternata* in vitro and in vivo. For those purposes, we examined numerous plant development and health metrics to estimate the agricultural plant-protecting potential of these biosurfactants. By completing these investigations, we intend to add to the expanding body of knowledge about biosurfactants’ production and possible uses in sustainable agriculture.

## 2. Materials and Methods

### 2.1. Bacteria

The *Bacillus* spp. used in this study were *Geobacillus stearothermophilus* DSM2313 purchased from Leibniz Institute DSMZ-German Collection of Microorganisms and Cell Cultures GmbH, *Bacillus subtilis* DSM10, and *Bacillus licheniformis* DSM13 acquired from the National Collection of Agricultural and Industrial Microorganism (NCAIM, Budapest, Hungary) under the identification number of, B.02624^T^ and B.02069^T^, respectively. To maintain strains, they were stored at 4 °C on Luria–Bertani (LB) agar plates (10.0 g/L tryptone, 5.0 g/L yeast extract, 10 g/L NaCl, 15.0 g/L agar).

### 2.2. Pathogenic Fungi

The pathogenic fungi used in the experiments were *Alternaria alternata* F.00969 purchased from NCAIM.

### 2.3. Fermentation

The fermentation was carried out in Biostat Q (B. Braun, Melsungen, Germany) bench-top fermenters with magnetic stirrers (300 rpm) temperature control, pH and oxygen electrodes (Mettler, Switzerland) additionally including a Hamilton Incyte living cell sensor and equipped with a cyclone for separating foams from the air outlet. For the biosurfactant fermentation, a minimal medium was used; 1 litre of minimal media contained 34.0 g glucose (Hungrana Kft., Szabadegyháza, Hungary), 6.0 g KH_2_PO_4_, 2.7 g Na_2_HPO_4_, 1.0 g NH_4_NO_3_, 0.1 g MgSO_4_ * ·7H_2_O, 1.2 × 10^−3^ g CaCl_2_, 1.65 × 10^−3^ g FeSO_4_*7H_2_O, 1.5 × 10^−3^ g MnSO_4_*4H_2_O and 2.2 × 10^−3^ g Na-EDTA [24,25]. In this investigation, the cell density in the culture was measured using the Hamilton Incyte in-line viable cell density (VCD) monitoring system. The conductivity signal of the Incyte sensor was discovered to be associated with pH. In the case of *Geobacillus,* conductivity also correlated to the product amount making possible real-time product measurements [13]. The system is comprised of three components: Incyte viable cell density unit DN12 capacitance sensor, a pre-amp signal transmitter, and a biomass controller, and all these parts are manufactured by Hamilton in Bonaduz, Switzerland.

### 2.4. Isolation of Biosurfactant

First, cells were removed from the fermented broth by centrifuging at 4000 rpm for 20 min with Janetzki K23D (MLW, GDR) set to 4 °C. The biosurfactants were then recovered using acid precipitation from the clear supernatant. The pH of the supernatant was set to 2 using 6 M HCl, and the temperature was maintained at 4 °C overnight. This precipitated the biosurfactants out of the solution, which was then centrifuged at 4000 rpm for 20 min to separate the biosurfactant-containing precipitate from the supernatant. The supernatant was dumped out, and the residue was resuspended. Using NaOH, the pH of the resuspended material was adjusted to 7. After pipetting the suspension into pre-weighed vials, it was lyophilised (Alpha 2-4, Martin Christ, Germany). We can determine the dry mass of the sample by calculating the difference between these two values, which provides us with the mass of the biosurfactant left after lyophilization [26,27,28].

### 2.5. Antifungal Assay

Biosurfactant’s antifungal activities were in vitro investigated utilising the poisoned food technique. Mixing 39 g of potato dextrose agar (PDA) powder with 1 litre of distilled water was followed by boiling, mixing, and autoclaving for 15 min until homogenous. After semi-cooling, various concentrations (0, 5, 10, 20, and 30%) of cell-free fermentation broth supernatant (1 mL/plate) containing biosurfactant were gently mixed with 25 mL PDA and put in a sterile petri dish before being further cooled. After solidification, a sterilised cork borer was transferred to a 6 mm agar disk from *A. alternata* culture grown for 7 days onto each culture plate. The cultures were incubated for an additional week at 25 °C. The diameter of the fungal colony was determined, and the inhibition percentage of the mycelial growth of the test fungus by the biosurfactant was calculated using the following formula [29].
(1)$$\mathrm{Inhibition}\;\mathrm{of}\;\mathrm{mycelial}\;\mathrm{growth}\;\left(\%\right)=\frac{\mathrm{dc}-\;\mathrm{dt}}{\mathrm{dc}}\times 100$$
where

dc represents the diameter of the fungal colony in the control group (without biosurfactant), anddt represents the diameter of the fungal colony in the treatment group (with the biosurfactant).

### 2.6. Planting, Transplanting, Growing, and Treating Plants

The planting of pinto beans (*Phaseolus vulgaris*) in 48 planting trays was performed with BIORG MIX potting soil. The soil was regularly watered and then covered with a transparent covering to minimise moisture loss. After one week, the seedlings were transferred into pots in triplicate, resulting in 10 groups of 3 plants each (30 plants in total). The plants were stored in an ozone-sterilised (UV-C lamp 2.5 W/2 m^2^, Anco Electrics Ltd.,Budapest, Hungary) 900 × 900 × 780 mm container. The plants were exposed to 3700 lux of light for 14 h daily at 25 degrees Celsius. The vegetation was watered three times each week. Using a spray bottle, infections and treatments were administered outside the box. The box was divided into two compartments by a transparent plexi sheet, one for control (not infected) and the other for fungal-infected plants. Plants without fungal infection consisted of nontreated (control) and bacterial broth treated ones (BS-0, GS-0, BL-0). *A. alternata* spores infected plants were divided into two groups, bacterial broth treatment before fungal infections (BS-A, GS-A, BL-A) and treatment after infection (A-BS, A-GS, A-BL) as indicated in Table 1. Thus, we could test the efficacy of both preventive (pre-infection treatment) and therapy (post-infection treatment) applications. Bacterial broth for treatment was made by fermenting *bacilli* for four days, centrifuging them, and the supernatant was sprayed onto plant leaves.

### 2.7. Disease Incidence and Severity

The disease severity index (DSI%) formula is a mathematical calculation that measures the severity of a disease in a particular population or sample. It is often used in plant pathology. The experiment was stopped at week eight after transplanting from the nursery. The incidence and severity of *Alternaria* infection on the leaves of the plants were determined according to the following classification shown in Table 2 [30].

Leaves with a value other than 0 were considered infected; the number of infected leaves divided by the total number of leaves was calculated as the incidence:(2)$$\mathrm{Disease}\;\mathrm{incidence}\;\left(\%\right)=\frac{\mathrm{number}\;\mathrm{of}\;\mathrm{infected}\;\mathrm{leaves}\;\left(-\right)}{\mathrm{total}\;\mathrm{number}\;\mathrm{of}\;\mathrm{leaves}\;\mathrm{on}\;\mathrm{plant}\;\left(-\right)}\times 100$$

The severity of the infection was calculated using the following formula:(3)$$\mathrm{Disease}\;\mathrm{severy}\;\mathrm{index}\;\left(\mathrm{DSI}\%\right):\frac{0\times n_{0}+1\times n_{1}+2\times n_{2}+3\times n_{3}+4\times n_{4}+5\times n_{5}}{5\times \sum n}\times 100$$
where

n_0_–n_5_ is the number of leaves in corresponding categories.

### 2.8. Chlorophyll Concentration of Leaves

The chlorophyll content of leaves was measured using a spectrophotometer. Each plant had a leaf of comparable size cut, weighed on an analytical scale, and then crushed with acetone to extract pigments. The pigments were then separated from the solvent by centrifuging the acetone solution in 15 mL Falcon tubes at 4000 rpm for 10 min. At two distinct wavelengths (=663, 645 nm) [31], the absorbance of the supernatant was determined using a spectrophotometer. Three repetitions of the measurement were performed using glass cuvettes.

Using the following formulas, the chlorophyll a and b concentrations in the leaf were calculated:(4)chlorophyll a mgg tissue=12.63×−2.52×A645×V1000×W
(5)chlorophyll b mgg tissue=20.47×A663−4.73×A645×V1000×W
(6)total chlorophyll mgg tissue=chlorophyll a+chlorophyll b
where

A663 is the absorbance of the solution measured at λ = 663 nm (-),

A645 is the absorbance of the solution measured at λ = 645 nm (-),

V is the volume of the solution (mL), and

W is the weight of the leaf (g).

### 2.9. Data Analysis

In this study, the means of the three treatments were compared using the Tukey test on all measurements. The statistical analysis was conducted utilising the IBM SPSS Statistics 26.0 software package (IBM Corp., Chicago, IL, USA) for data evaluation and analysis. To determine statistical significance, the significance level for all statistical tests was set to *p* < 0.05.

## 3. Results

### 3.1. Fermentation

Figure 1 introduces the time course of the fermentation. The usefulness of the in-line sensor is evident since the significant cell growth occurred in samplings gaps, i.e., during the night. All bacteria reached an OD600 value between 2.5–3, while 1.1–1.3 g/L could be reached regarding product amounts. *B. licheniformis* was found as most productive since it reached a plateau in 10 h compared to the other two bacteria around 20 h. The time courses also demonstrate the correlations of permittivity-based VCD measurements to off-line determined OD600 values and conductivity to pH values.

#### Verification of In-Line Sensor Usefulness

Correlations are more exactly presented in Figure 2, where VCD is plotted versus OD600, resulting in correlation coefficients between 0.6 and 0.92, where the lowest value belongs to *B. subtilis.* The lack of off-line determined middle-range data for those bacteria because of the night period causes a bit higher uncertainty besides an acceptable correlation coefficient.

Figure 2 depicts that the correlation between VCD and OD600 has a substantial positive link, especially for *B. licheniformis* and *G. stearothermophilus,* with an R^2^ of 0.9262 and 0.9202, respectively. This indicated that as permittivity rises, so does OD600.

In Figure 3, the correlation between conductivity and pH with an R^2^ value of 0.9043, 0.889, and 0.6363 is observed for *B. licheniformis, B. subtilis,* and *G. stearothermophilus*, respectively. The weak correlation for the latter is probably because of an observed time shift between pH and conductivity recording.

To summarise the usefulness of the in-line sensor, one can conclude that out of the tested 6 correlations, 4 were close to or over 0.9, indicating a strong correlation between both permittivity-based VCD and optical density as well as pH and conductivity. The outstanding of *G. stearothermophilus* can be because of the different behaviour of its biosurfactant. It was reported that this strain produces a polymer-type biosurfactant instead, while the other two produce lipopeptide type biosurfactants [31].

### 3.2. Antifungal Assay

The data provided (Table 3.) are the findings of an antifungal experiment using the poisoned food technique to determine the impact of varying doses of *B. licheniformis*, *B. subtilis*, and *G. stearothermophilus* on the development of mycelium in a petri dish, i.e., in vitro.

Higher percentages indicate in Table 4. more suppression of mycelial development. The average growth of the control group was 89.68 ± 2.19 mm without any inhibition. The findings suggest that all three bacterial strains inhibited mycelial growth, with the most significant suppression (above 70%) reported at the highest dose tested (30%) for all *Bacillus* spp.—fermented broth.

### 3.3. Disease Incidence and Severity

Fungal disease occurrence (Figure 4.) was significantly lower in the case of all three tested bacterial supernatants if applied before fungal infection. Post-infection treatments are inefficient since disease occurrences were similar to untreated but infected controls.

However, Figure 5 depicts the impact of the three tested biosurfactants on the severity of the fungal infections, which is remarkably lower in the case of *B. subtilis* and *B. licheniformis* but is higher for *G. stearothermophilus*. At the same time, pre-infection treatment was observed to be more efficient than post-infection treatment. These indicate that emulsifying biosurfactants (polymeric type) are less effective against fungal infection than surface-tension-reducing (lipopeptide type).

### 3.4. Chlorophyll Concentration

The decrement in chlorophyll contents (Figure 6.) indicates a decrease in the photosynthetic activities of the tested plants. The nontreated and non-infected control group contained one of the highest concentrations of chlorophyll a and b, suggesting that its photosynthetic activity is the highest. Similar chlorophyll contents were found only for *bacilli-treated* groups (BS-0, GS-0, BL-0), which indicates that these bacterial treatments are not harmful to bean plants. The statistical analysis reveals significant differences in chlorophyll levels across the groups, as shown by the values of Pr > F. (all less than 0.001). The lowest chlorophyll contents could be obtained in the post-infection treatment group, which confirms our above-described findings, i.e., pre-infection bacterial supernatant treatment is more efficient, resulting in 90% chlorophyll content of uninfected plants.

### 3.5. Effect of Treatment and Infections on Plant Height

Figure 7 demonstrates that changes in total plant height, root length, and shoot length are not significant; thus, this method is not sensitive enough to indicate the fungal disease and its bacterial treatment. The ability of *A. alternata* to display both endophytic and pathogenic activity exemplifies this fungus’ adaptability and its ability to interact with a diverse array of host plants. This diversity may be attributable, in part, to the wide range of metabolites generated by *A. alternata* and the taxonomic group to which it belongs, which might support a variety of host interactions and nutritional strategies [32,33]. Somewhat higher lengths were obtained for *B. subtilis* treated but not infected plants. These might indicate that *B. subtilis*-based antifungal treatments may positively affect plant height, chlorophyll contents, disease severity and occurrences, and inhibition of fungal growth in vitro. While the Pr > F values are more than 0.05, statistical analysis reveals that none of the variations in plant height between the groups was statistically significant.

## 4. Discussion

The study explored the fermentation of biosurfactants by *B. licheniformis* and discovered a good correlation between the conductivity and primary fermentation products and between permittivity and the optical density of the fermentation broth. Such *in-line* real-time and continuous measurements support a better understanding of biosurfactant formation and reach higher productivity via cutting fermentations in optimal time [34,35]. The comparison of the tested three potential bio-fungicide-producing bacteria from fermentation revealed that the fastest cell growth could be reached by *B. licheniformis* with the highest biosurfactant amount resulting in the highest productivity. Compared to *B. subtilis* and *G. stearothermophilus*, *B. licheniformis* generated a larger output of biosurfactant throughout the fermentation process, with 15% and 5.2% more crude product, respectively. Compared to earlier research, which has reported yields ranging from 0.5 g/L to 1.8 g/L [3,36,37], the yield of the present study is relatively high. Our results indicate that our biosurfactant production method effectively achieved a high crude product yield.

The investigation into the potential of three bacterial strains to inhibiting the plant pathogenic fungus of *A. alternata* indicated that all of them were efficient under in vitro test conditions. However, in vivo plant experiments showed that the *G. stearothermophylus* broth supernatant was less successful in reducing disease incidence and severity. In contrast, the plants’ height and chlorophyll content were similar to the other bacterial treatments. On the other hand, it was discovered that *B. subtilis* and *B. licheniformis* were equally successful in treating the plants regarding severity and incidence. It is hypothesised that variations in bacterial biosurfactants may account for the different degrees of efficacy found in in vivo experiments. Including lipopeptide-type biosurfactants, such as lichenysin by *B. licheniformis* and surfactin by *B. subtilis,* in the broth resulted in comparable efficacy; however, polymeric-type emulsifying biosurfactants in the supernatant of *G. stearothermophilus* [31] resulted in a decreased efficacy.

The biosurfactant-producing bacteria *B. subtilis*, *G. stearothermophilus*, and *B. licheniformis* reduced the severity of *A. alternata* disease by up to 30%, 5%, and 25%, respectively, when applied to plants (i.e., in vivo). Additionally, the chlorophyll a and b concentrations of the bacterial supernatant-treated plants ranged from moderate to high, suggesting enhanced photosynthetic activity.

Overall, the research indicates that *B. licheniformis* is the most efficient bacterium for biosurfactant generation and biosurfactant treatment of plants. This work adds to the expanding body of knowledge on the use of biosurfactants for sustainable agriculture and indicates the potential for more research and development in this area.

## 5. Conclusions

The correlation coefficients between conductivity and pH and the association between permittivity and OD600 imply that these variables might be utilised to monitor biosurfactant production on a wide scale.

In conclusion, we can observe that lipopeptide-type biosurfactants are more effective bio fungicides than polymeric-type emulsifiers. The reason can be that air-mycelia and conidiospores (having somewhat hydrophobic behaviour) of plant pathogenic filamentous fungi can be better attacked if surface tension is controlled by biosurfactant-based bio fungicide in comparison to emulsification. The tested biosurfactant-producing bacilli, namely *B. licheniformis* and *B. subtilis,* are frequently used in biotechnology for different purposes, where the bacterial biomass is a discarded by-product. Our study highlighted the application of these cells as bio-fungicide, which can be a sustainable solution in the future.

Additionally, regarding in vivo plant treatments, increasing the sample size and replicating could help reduce the variability in the results and increase the study’s statistical power. Improved disease severity analyses using digital imagery and the connection between biosurfactant treatment and plant yield can be an interesting continuation.

## Figures and Tables

**Figure 1 microorganisms-11-00707-f001:**
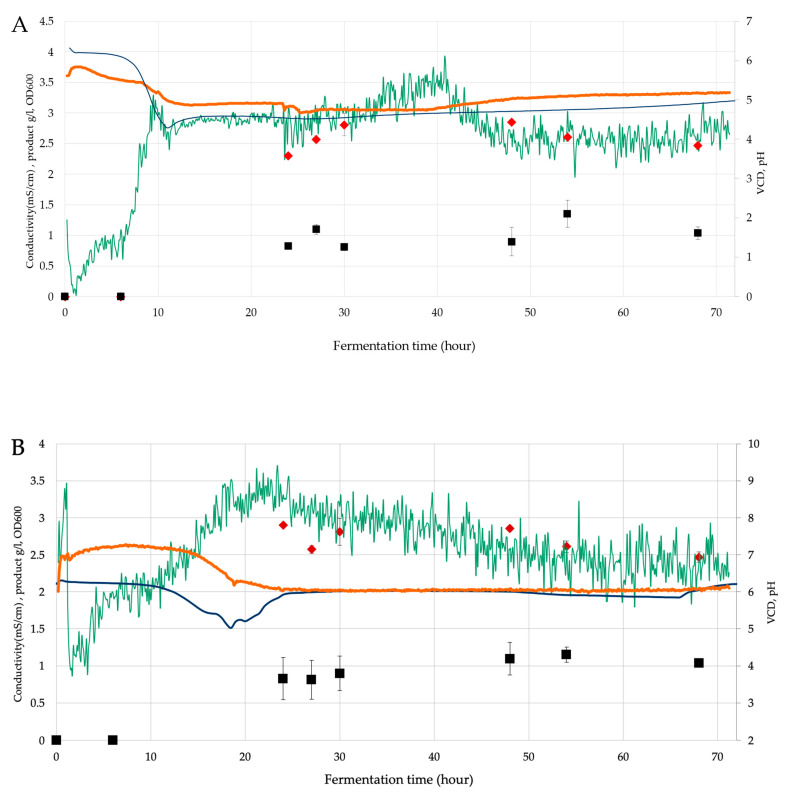
Time course of biosurfactant fermentations: (**A**) *B. licheniformis* ▬ *In-line* detected total products IDTP (=0.65 × conductivity [mS/cm] + 0.1); (**B**) *B. subtilis* ▬ IDTP (=0.65× conductivity [mS/cm] + 0.1); (**C**) *G. stearothermophilus* ▬ IDTP (=0.65* conductivity [mS/cm] + 0.1) biosurfactant fermentation, ▬ pH, ▬ VCD (=10× permittivity [mS/cm] + 1.8), ◊ OD600 and ■ product.

**Figure 2 microorganisms-11-00707-f002:**
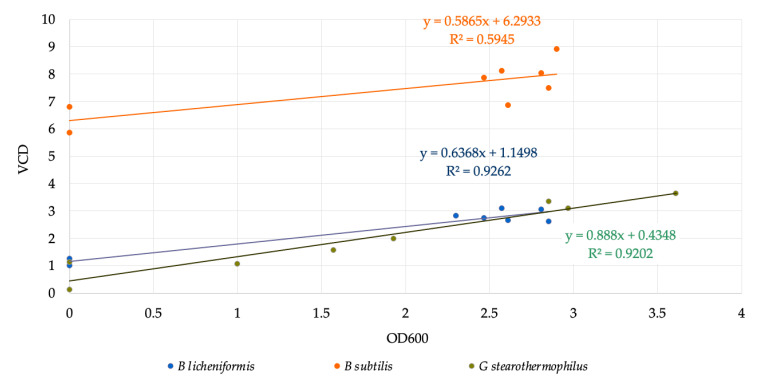
Correlation curve for VCD and OD600.

**Figure 3 microorganisms-11-00707-f003:**
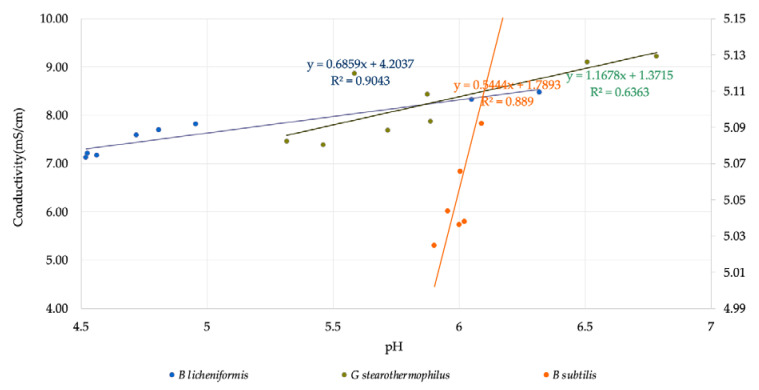
Correlation curve for conductivity and pH.

**Figure 4 microorganisms-11-00707-f004:**
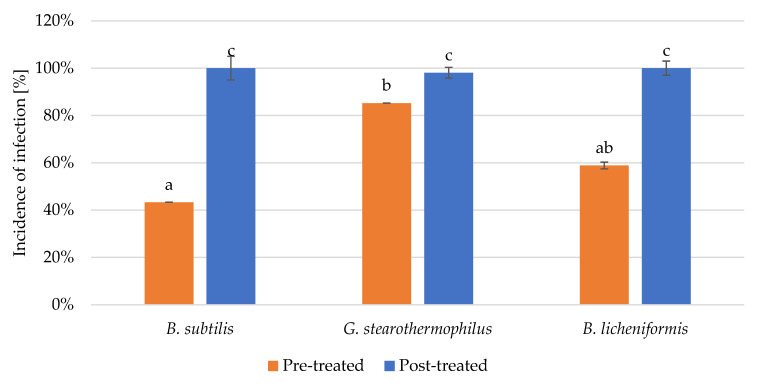
Disease incidence of *A. alternanta* pre-infection treatment and post-infection treatment by *Bacilli* supernatants comparing mean significance with Tukey HSD. Small letters indicate the statistically different results groups.

**Figure 5 microorganisms-11-00707-f005:**
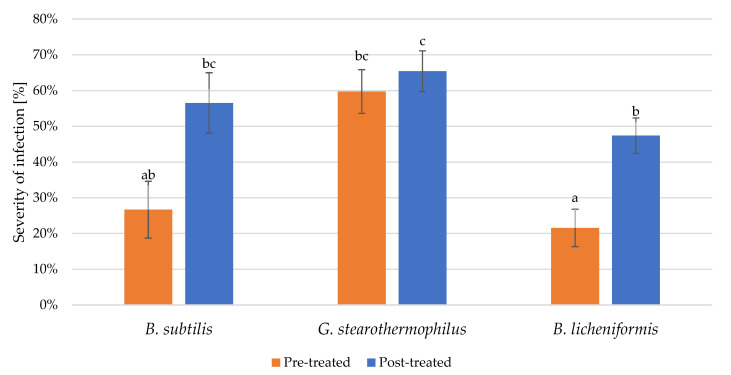
Disease severity of *A. alternanta* pre-treated and post-treated by biosurfactant comparing mean significance with Tukey HSD. Small letters indicate the statistically different results groups.

**Figure 6 microorganisms-11-00707-f006:**
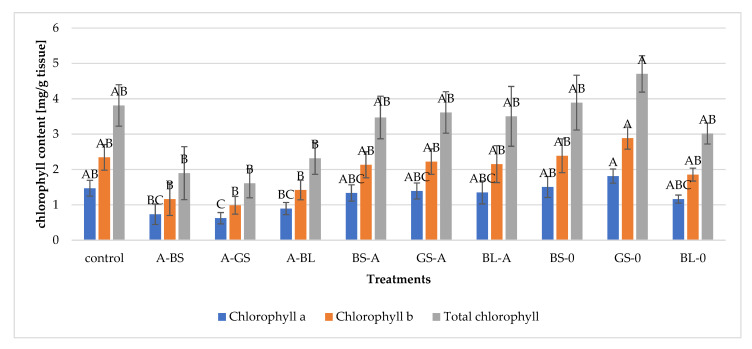
Results of biosurfactant treatments and fungal infections on the chlorophyll content comparing mean significance with Tukey HSD of bean’s leaves: BS-A: pre-treatment with *B. subtilis* before *A. alternata* infection; GS-A: pre-treatment with *B.subtilis* before *A. alternata* infection; and BL-A: pre-treatment with *B. licheniformis* before *A.alternata* infection; A-BS, A-GS, and A-BL: pre-treatments with supernatant of *B. subtilis*, *G. stearothermophilus,* and *B. licheniformis* supernatant; uninfected but *B. subtilis*, *G. stearothermophilus* and *B. licheniformis* treated plants. Capital letters indicate the statistically different results groups.

**Figure 7 microorganisms-11-00707-f007:**
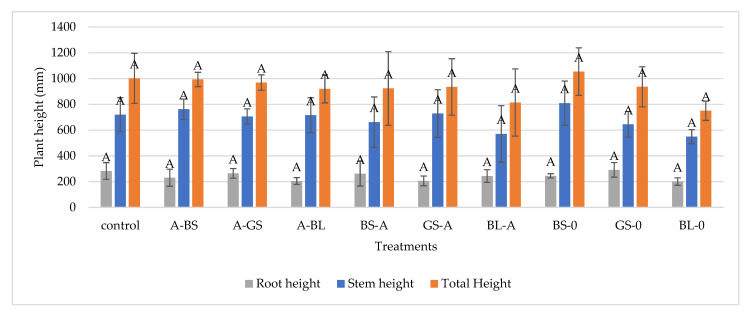
Effect of different treatments on plant heights comparing mean significance with Tukey HSD. Capital letters indicate the statistically different results groups.

**Table 1 microorganisms-11-00707-t001:** Configurations of plant treatments.

	Pre-Treatment (Before *Alternaria* Infection)	Post-Treatment (After *Alternaria* Infection)	Non-Treatment
*B. licheniformis*	BL-A	A-BL	BL-0
*B. subtilis*	BS-A	A-BS	BS-0
*G. stearothermophilus*	GS-A	A-GS	GS-0

**Table 2 microorganisms-11-00707-t002:** Score for disease severity rating given to the plant leaves.

Extent of Infection	Score
infection	0
less than 1% of leaf infected	1
1–10% of leaf infected	2
11–25% of leaf infected	3
more than 25% of leaves infected, leaves still green	4
more than 25% of leaf-infected, leaf dead	5

**Table 3 microorganisms-11-00707-t003:** Inhibition of *A. alternata* by *B. licheniformis* at 5% (A); 10% (B); 20% (C); and 30% (D); *B. subtilis* at 5% (E); 10% (F); 20% (G); and 30% (H); and *G. stearothermophilus* at 5% (I); 10% (J); 20% (K); and 30% (L); relative to the control (M).

	5%	10%	20%	30%
*B. licheniformis*	A 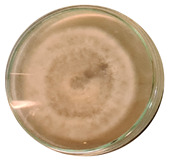	B 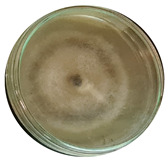	C 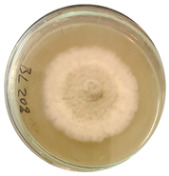	D 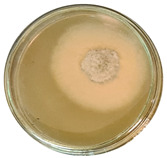
*B. subtilis*	E 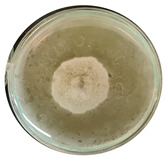	F 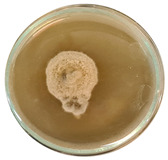	G 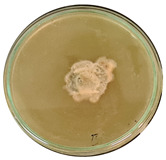	H 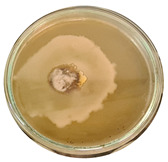
*G. stearothermophilus*	I 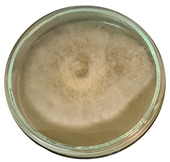	J 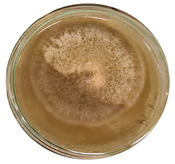	K 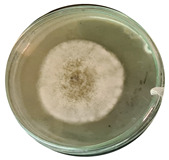	L 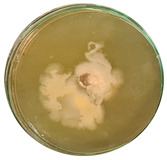
Control	M 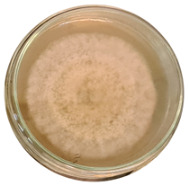			

**Table 4 microorganisms-11-00707-t004:** In vitro inhibition of mycelial growth (%).

Treatment	Concentration	Mean Average Growth of Mycelium (mm)	Inhibition of Mycelial Growth (%)
*B. licheniformis*	5%	65.45 ± 4.24	27.01
	10%	59.89 ± 9.90	33.22
	20%	55.38 ± 7.07	38.24
	30%	23.23 ± 5.66	74.09
*B. subtilis*	5%	32.84 ± 2.51	63.38
	10%	29.13 ± 4.95	67.51
	20%	23.22 ± 4.72	74.10
	30%	19.92 ± 9.19	77.78
*G. stearothermophilus*	5%	67.74 ± 8.03	24.46
	10%	63.81 ± 8.08	28.84
	20%	47.68 ± 5.95	46.83
	30%	18 ± 8.99	79.93
Control		89.67 ± 2.19	0.00

## Data Availability

The data presented in this study are available on request from the corresponding author.

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
