# Peer review of "The Potential of Bacilli-Derived Biosurfactants as an Additive for Biocontrol against Alternaria alternata Plant Pathogenic Fungi"

_microorganisms, 2023, doi:10.3390/microorganisms11030707_

Round 1
Reviewer 1 Report
An overview scientific merit of the study is credible and is of interest, but the flow of information have limitations to follow.
The abstract of the manuscript is well written with greater clarity which catches interest.
General comment: the overall manuscript consist of typos and inconsistance on the use of scientific names as indicated in the attached pdf with comments.
The introduction of the presented manuscript also gave a good background to the study. However, Materials and Methods section needs to be re-written with more clarity for easy follow-up.
Results section and discussion sections are similar except given different headings. These two sections should be distinct; the result sections should report the results of the study, and the discussion section give detailed relevance of the results of the study and novelty.
Conclusion section also need revision.

Author Response
Dear Reviewer,
We appreciate you taking the time to look over our manuscript. We have made changes based on what you recommended us and have carefully thought about all your suggestions for improving the manuscript. Here are all the changes made and the answers to the questions you asked.
- An overview scientific merit of the study is credible and is of interest, but the flow of information have limitations to follow.
- The abstract of the manuscript is well written with greater clarity which catches interest.
- General comment: the overall manuscript consists of typos and inconsistence on the use of scientific names as indicated in the attached pdf with comments.
The adequate changes have been made and the typos and inconsistences in the use of scientific names has been fixed.
- The introduction of the presented manuscript also gave a good background to the study. However, Materials and Methods section needs to be re-written with more clarity for easy follow-up.
The materials and method part has been rewritten and supplemented for better understanding.
- Results section and discussion sections are similar except given different headings. These two sections should be distinct; the result sections should report the results of the study, and the discussion section give detailed relevance of the results of the study and novelty.
- Conclusion section also need revision.
Results, Discussion and Conclusion sections were thoroughly rewritten according to kind recommendation of the Reviewer.
Thank You again for Your efforts making our manuscript improved!

Reviewer 2 Report
The manuscript entitled "The potential of bacilli-derived biosurfactants as an additive for biocontrol against Alternaria alternata plant pathogenic fungi" requires critical revision with due consideration on the following issues:
1. The manuscript has multiple spelling mistakes and syntax errors which have been duly marked with orange lines in the reviewer’s attachment. The scientific names of the microorganisms also need to be in italics, throughout the MS. Please take due care to revise the MS critically.
2. Figures 4, 5, 6, 7 the significant differences among group means have not been calculated. The MS presents a preliminary study and over that if the results presented do not report statistical significance, it would not validate the credibility of the data.
3. Materials and Methods:
sub-heading 2.6. Planting, transplanting, growing, and treating of plants:
does not present any information on variety of common beans, neither has any information on the methodology used for seed treatment with bacterial isolates has been presented. Kindly include detailed description of the same.
Also, please tabulate all the treatments with their descriptions at the end of this section.
4. Reporting plant height without presenting correlating data of plant fresh/ dry weights is inconclusive.
5. There is hardly any discussion on the huge SD values obtained in the results of most treatments.
6. Please include the future prospects of this study in the conclusion section.

Author Response
Dear Reviewer,
We appreciate you taking the time to look over our manuscript. We have made changes based on what you recommended us and have carefully thought about all your suggestions for improving the manuscript. Here are all the changes made and the answers to the questions you asked.
The manuscript entitled "The potential of bacilli-derived biosurfactants as an additive for biocontrol against Alternaria alternata plant pathogenic fungi"
- The manuscript has multiple spelling mistakes and syntax errors which have been duly marked with orange lines in the reviewer’s attachment. The scientific names of the microorganisms also need to be in italics, throughout the MS. Please take due care to revise the MS critically.
Yes, we affirm that all essential adjustments have been made to the text, including spelling errors, syntax errors, and the usage of italics for scientific names of microorganisms. We have meticulously revised the material to guarantee its correctness and clarity.
- Figures 4, 5, 6, 7 the significant difference among group means have not been calculated. The MS presents a preliminary study and over that if the results presented do not report statistical significance, it would not validate the credibility of the data.
Thank You for the remark, with which we agree. Therefore, we supplemented the manuscript with the details and results of statistical analysis including significance between results groups. These confirms our described findings and enhanced the quality of the manuscript.
- Materials and Methods:
sub-heading 2.6. Planting, transplanting, growing, and treating of plants: does not present any information on variety of common beans, neither has any information on the methodology used for seed treatment with bacterial isolates has been presented. Kindly include detailed description of the same. Also, please tabulate all the treatments with their descriptions at the end of this section.
We are very grateful for this comment, thus the section 2.6 on planting, transplanting, cultivating, and treating plants has been supplemented with the details. Particularly, we have given a description of the variety of common beans utilised in our research, as well as the process for treating seeds with bacterial isolates. In addition, we have created a table at the end of section 2.6 (which we also present here) that summarises all of the treatments used in our research, as well as their descriptions simplifying the understanding.
|
|
Pre-treatment |
Post-treatment |
Non-treatment |
|
B. licheniformis |
BL-A |
A-BL |
BL-0 |
|
B. subtilis |
BS-A |
A-BS |
BS-0 |
|
G. stearothermophilus |
GS-A |
A-GS |
GS-0 |
- Reporting plant height without presenting correlating data of plant fresh/ dry weights is inconclusive.
We agree with the Reviewer, that plant weight is also an important growth indicator. However, we intended to supply homogenous illumination conditions, therefore we assumed, that plant height is correlating with plant weights. Additionally, other similar studies also reported plant height without weight as growth indicator for example M.S.Attia et al [1]
- There is hardly any discussion on the huge SD values obtained in the results of most treatments.
Remarkably high SD values were observed and presented regarding plant height data.
For small scale plant treatment experiments, similar high SD values are frequently reported [2],[3] . For better reliability and smaller SD values much higher involved plant numbers are required like field experiments. Therefore, in conclusion we implemented the following sentence: “Regarding in vivo plant treatments increasing the sample size and replicating could help to reduce the variability in the results and increase the statistical power of the study.”
- Please include the prospects of this study in the conclusion section.
Thank You for the suggestion, we involved the followings there:
“The tested biosurfactant producing bacilli namely B. licheniformis and B. subtilis are frequently used organism in biotechnology for different purposes, where the bacterial biomass is a discarded by-product. Our study highlighted the application of these cells as bio-fungicide, which can be a sustainable solution in the future.”
Authors are very grateful for receiving these recommendations for improving their manuscript!
[1] https://link.springer.com/article/10.1007/s12010-022-03975-9
[2] https://bmcplantbiol.biomedcentral.com/articles/10.1186/1471-2229-14-25
[3] https://link.springer.com/article/10.1007/s11258-019-00936-x

Round 2
Reviewer 1 Report
The manuscript has significantly improved.
Author Response
Authors are very grateful for Reviewer's efforts in improving the quality of our manuscript. We are also thankful for the last remark as "The manuscript has significantly improved."

Reviewer 2 Report
The MS has been significantly revised and all the previous comments and suggestions were duly taken into consideration but following two points require attention:
1. Figures 6 and 7: SD values applied only on total chlorophyll and total height, respectively. Please apply the same for all treatments. Also in figure 7, x-axis please include the unit.
2. The MS still requires significant language correction.
Author Response
The MS has been significantly revised and all the previous comments and suggestions were duly taken into consideration but following two points require attention:
- Figures 6 and 7: SD values applied only on total chlorophyll and total height, respectively. Please apply the same for all treatments. Also in figure 7, x-axis please include the unit.
Thank You for the remark, we added also that SD values to Fig.6., and also the missing unit on Fig.7. y-axis was placed.
- The MS still requires significant language correction.
Thank You for this remark. In such a short revision time we tried to improve the lingual level of the manuscript with the help of Grammarly. Its suggestions were supervised and most of them were accepted.
Authors want to express their greatest appreciation to Reviewers effort improving our manuscript.
